# Intensive care nurse managers' experiences during the first wave of the Covid-19 pandemic: Implications for future epidemiological crises

Beata Dobrowolska[1]*, Aleksandra Gutysz-Wojnicka[2], Magdalena Dziurka[1], Patrycja Ozdoba[1], Dorota Ozga[3], Beata Penar-Zadarko[4], Renata Markiewicz[5], Agnieszka Markiewicz-Gospodarek[6], Alvisa Palese[7]

1 Department of Holistic Care and Nursing Management, Faculty of Health Sciences, Medical University of Lublin, Lublin, Poland, 2 Department of Nursing, School of Health Sciences, Collegium Medicum, University of Warmia and Mazury in Olsztyn, Olsztyn, Poland, 3 Institute of Health Sciences, College of Medical Sciences of the University of Rzeszow, Rzeszow, Poland, 4 Department of Nursing and Public Health, Laboratory of Methodology of Research and Education in Nursing, Institute of Health Sciences, College of Medical Sciences of the University of Rzeszow, Rzeszow, Poland, 5 Department of Neurology, Neurological and Psychiatric Nursing, Medical University of Lublin, Lublin, Poland, 6 Department of Human Anatomy, Medical University of Lublin, Lublin, Poland, 7 Department of Medical Sciences, University of Udine, Udine, Italy

* beata.dobrowolska@umlub.pl

## Abstract

### Background

Nurse managers play an important role in coordinating the multidisciplinary teamwork, which is specifically important in emergency and crises situations like the COVID-19 pandemic. The aim of this qualitative study is twofold: (1) to explore the experiences of the Intensive care units (ICU) nurse managers regarding their work during the first wave of the COVID-19 pandemic, and (2) to analyse what implications might be provided based on experiences of nurse managers for future possible epidemiological crises.

### Methods

In-depth phone interviews were conducted to explore the experiences of ward managers–nurses (n = 15) working in different hospitals across Poland. Interviews were taped and transcribed verbatim, and then qualitatively analysed.

### Results

Three main categories were identified: *(1)* Challenge of working with the unknown, *(2)* Nurse managers' expectations, and *(3)* Methods of coping and received support. The COVID-19 pandemic strongly affected the work of ICU nurse managers and uncovered the malfunctioning of the healthcare system.

**Data Availability Statement:** Data (transcript of interviews) contain potentially identifying and sensitive ICU nurse managers information

therefore it cannot be shared. This fact is restricted by the Main Board of the Polish Association of Anesthesia and Intensive Care Nurses, which approved protocol of the study, acting according to the European Union General Data Protection Regulation. Data is available upon request in the Chair of Integrated Nursing Care at the Medical University of Lublin where the data is stored using email address: katedra.zop@umlub.pl.

**Funding:** The authors received no specific funding for this work. The funders had no role in study design, data collection and analysis, decision to publish, or preparation of the manuscript.

**Competing interests:** The authors have declared that no competing interests exist.

## Conclusion

It is important to improve the knowledge and competence of hospital management personnel through exercises and in-service training on how to handle emergencies in order to improve the management of healthcare facilities, increase the safety of patients and employees, and the quality of healthcare.

## Introduction

The coronavirus pandemic suddenly paralysed the entire world, as less than a two months after Coronavirus Disease 2019 (COVID-19) was detected, the World Health Organization declared the contagious disease an international threat to public health [1, 2]. Nurses were on the front line of the fight against this dangerous virus, with an increased risk of contracting the coronavirus and becoming infected with COVID-19 [3]. This sudden situation has turned the operation of hospitals upside down and, above all, forced numerous changes in the working environment for nurses [4].

The sources of additional burdens for nursing staff included the said organizational changes, isolation, excessive workload, and a lack of or insufficient facilities in the form of appropriate equipment, including personal protective equipment (PPE) [5]. Furthermore, wearing protective clothing for 8–12 hours while on duty caused discomfort and dehydration, and some types of masks, such as N95, after being used for extended periods of time, resulted in abrasions of the facial epidermis [6]. All these factors contributed to physical, but also mental exhaustion among the nursing staff [7].

The dynamically changing situation from one day to another led to misinformation among healthcare workers, including nurses. It was problematic to find one's way around quickly changing procedure patterns, leading to feelings of helplessness among the medical personnel [8]. The main source of fear during the outbreak among nurses stemmed from the possibility of contracting or carrying Severe Acute Respiratory Syndrome-Coronavirus 2 (Sars-CoV-2)– they feared for their own health, as well as the health of their co-workers and family members [9]. Many scientific publications addressed the emotional states experienced by the nurses while battling coronavirus. Unfortunately, this negatively affected their mental health, generating high levels of stress, anger, anxiety, and helplessness, which caused professional burnout among nursing staff [10–14].

During such epidemiological crises the crucial role is played by nurse managers. They take the responsibility for the atmosphere at work, the safety and well-being of their nurse colleagues, and the organization and distribution of tasks, which ultimately translates into the quality of patient care [15]. In the situation of the COVID-19 pandemic, nurse managers have the responsibility to manage the crisis prudently, based on collaboration and data analysis, with empathy, honesty and nurturing supportive relationships to alleviate nurses' anxiety and sadness not only now, but in the years to come [16]. The survival of the system in times of crisis is determined, among other things, by how it is managed, which is why nurses who manage the entire unit play such an important role [17]. Nurse managers in hospitals face additional pressure in the workplace during times of crisis, when they should also be able to deal effectively with emerging problems and achieve the most promising results [18]. Previously, there had not been such a major health crisis, at least on such a scale, where nurse managers had to make extreme ethical decisions that touched on the allocation of insufficient resources [19].

Organizing a space where nurse managers feel supported and valued will not only enable them to find the right power to demonstrate courageous leadership in situations of uncertainty

but will also contribute to the organization's effectiveness in crisis situations [20]. Addressing such challenges and using coping strategies is highly dependent on the past experiences of nurse managers caused by the pandemic crisis [21]. Therefore, it seems reasonable to further explore this area and clarify the experiences of nurse managers during the coronavirus pandemic, which will allow more effective management of the emergency, pandemic, crisis situations in the future.

## Methods

### Aim

The aim of this qualitative study was twofold: (1) to explore the experiences of the ICU nurse managers regarding their work during the first wave of COVID-19 pandemic, and (2) to analyse what implications might be provided based on experiences of nurse managers for future possible epidemiological crises.

### Study design

Qualitative research design employing phone interview technic was carried out in Poland between May and June of 2020. Decision of choosing phone interviews was influenced by two main factors: firstly, it was the 1st wave of COVID-19 pandemic with strict lockdown and it was the only way to collect data referred to study aim; and secondly, phone interviews organised in suitable for nurse managers time helped them to freely share their experience and to provide a kind of relief when talking about their current situation with somebody from outside the hospital. To minimalise the risk of bias, the authors followed four criteria for trustworthiness created by Lincoln and Guba (1) Credibility, (2) Transferability, (3) Dependability, (4) Confirmability [22, 23]. Also, to increase rigor, triple coding of data during the analysis was used; determination of data saturation was performed; the researcher who conducted the interviews was not engaged in any relationship with nurse managers; the interviews were collected by only one researcher. The study is reported using COREQ checklist (Consolidated criteria for reporting qualitative research), (see S1 File), [24].

### Study participants and setting

The study involved 15 nurse managers working in hospitals across Poland. Purposeful sampling was used, taking into account intensive care units in hospitals of various references throughout Poland. After the 15th interview (or thereabouts), most of the same ideas had been repeatedly expressed. Therefore, the researchers felt that theoretical saturation was accomplished by the time the 15th interview was completed [25]. And 15th interviews were accepted for final analysis.

The inclusion criteria were: (1) nursing licence, (2) work as a nurse manager in intensive care units in the time of the first wave of the COVID-19 pandemic, (3) consent to participate in the study.

### Data collection process

Snowball sampling method was used to collect contacts to nurse managers. Interviews were conducted by only one female researcher (B.D.). Due to the pandemic and containment situation, interviews were held via telephone, with loudspeakers, to perceive different expressions, tones of voice, or silences.

First, each nurse manager gave their consent to share their phone number with the researcher. Then, a text message was sent to arrange the most convenient time for the nurse manager to take the interview. Once this had been agreed, the researcher called the respondent

and once again informed them about the study aim and process, and that the interview was going to be recorded. After obtaining verbal consent, the interview was recorded. Interviews lasted between 15.30 and 43.52 minutes. Each nurse manager was assigned a code (WM1, WM2, etc.) to ensure anonymity. The interviews were transcribed verbatim, with no changes to the context of the statements, using Microsoft Word 2017 software. After the interview, some nurse managers stressed that it was a kind of mental relief for them to talk about their current situation. In all phases of the research, triangulation of data and researchers was performed to improve the validity of the results and thus to acquire a greater understanding of the lived and studied reality. In data triangulation, researchers took care to collect phone interviews with ICU nurse managers from different regions of Poland and from different ICUs (number of beds and number of nursing staff managed). In researchers triangulation, three researchers analysed of transcribed text independently, then worked together discussing similarities and differences and choosing quotes for exemplification of identified categories and subcategories.

The interviews were guided using a combination of open-ended questions shown in Table 1.

## Data analysis

The analysed research material included 140 pages of transcribed text. Qualitative analysis was performed by three researchers. The Linsdeth and Norberg (2004) three-step, phenomenological-hermeneutic method was used [26]. The first of these involves "naive reading" of transcripts to determine the meaning of the data and includes reading the interview transcripts several times. The next step is a structural analysis, during which the researcher makes a preliminary identification and then formulation of categories and subcategories in everyday language to convey the meaning of the respondent's experience. The final step involves performing a critical, in-depth interpretation of the obtained data in relation to this data as a whole, followed by reflection and rethinking of the identified categories [26].

## Ethical considerations

The study protocol was approved by the Main Board of the Polish Association of Anesthesia and Intensive Care Nurses (Resolution No 21/VI/2020 dated 22.04.2020) and was prepared and done in line with the standards set forth in the Declaration of Helsinki [27].

**Table 1. Questions which guided the interview.**

| No | Questions |
|---|---|
| 1 | How long have you worked as a nurse? And how long have you worked as a nurse manager? |
| | How many nursing staff work in your unit and how many sites/beds are there in your unit? |
| 2 | What job-related problems are you currently experiencing most often in the workplace? |
| 3 | What does the collaboration with the nursing team, medical team and other medical professionals look like in your ward? Has anything changed in this regard? |
| 4 | What are the emotions that currently accompany you at work? What are your emotions at home? Has any of these emotions been dominating recently, and how do you deal with it? |
| 5 | What ethical dilemmas are you currently experiencing? |
| 6 | As a nurse manager, what actions do you expect to be taken by your supervisors? |
| 7 | In your opinion, who can be of help to you the most? What do you need most at the moment? As a nurse manager, where do you draw support from in your current situation? Who provides the most substantial and effective help? |
| 8 | What do you think about the future and the life after the pandemic? |
| 9 | Is there anything you would like to add that has not been mentioned but has recently been relevant to your work as a nurse manager? |

To ensure anonymity and confidentiality, the research material was assigned codes. The respondents' verbal consent was accepted and their phone numbers were deleted after participation. Verbal consent was obtained from the study participants several times. First time, when intensive care units nurse managers consented to give their phone numbers, then, when nurse managers answered to the researcher question about their availability for interview, and finally, before the starting and recording interview. When nurse managers agreed for being interviewed and recorded, recording interview started from consent to participate in the study statement to document this fact in a transcription. Additionally, the study protocol included detailed information about the process of obtaining verbal consent for study participation and as such was approved by the Main Board of the Polish Association of Anesthesia and Intensive Care Nurses. In the characteristics section, the mean values of their work experience were given to avoid their identification. The interviewer did not remain in any relationship with the nurse managers.

## Results

### Study participants

The study was participated in by 15 nurse managers (13 women, 2 men) from hospitals in different regions of Poland. Their average job seniority as a nurse was 29.73 years (SD = 7.81), and a nurse manager– 12.81 years (SD = 7.94), (Table 2). The number of sites/patient beds for which the respondents were responsible ranged from 5 to 23, and the number of personnel– from 17 to 203.

### Identified categories

Three main categories were identified: (1) *Challenge of working with the unknown* (with six subcategories: (a) Dealing with deficiencies, barriers and chaos, (b) New expectations of nursing workforce, (c) New roles of nurse managers, (d) Collaborating and dealing with different staff groups, (e) Emotions and the sense of responsibility for the team, (f) Current ethical dilemmas), (2) *Nurse managers' expectations* (with two subcategories: (a) Urgent needs and (b) Thinking about the future), (3) *Methods of coping and received support*.

**Challenge of working with the unknown.** *Dealing with deficiencies, barriers and chaos.* Some of the most frequently mentioned challenges during the first wave of the COVID-19 pandemic were as follows: (1) shortages of personal protective equipment, (2) barriers in hospital infrastructure, (3) shortages of medical personnel, (4) lack of clear operating procedures, and (5) insufficient knowledge of how to keep oneself and the patients safe.

Nurse managers reported difficulties in placing purchase orders for personal protective equipment. Usually, they were responsible for arranging and ensuring its availability. The staff had to work with equipment which was often obsolete. Looking for ways to overcome the

**Table 2. Selected variables of nurse managers (*n* = 15).**

| Variables | | Min. | Max. | M | SD |
|---|---|---|---|---|---|
| Years of service as a nurse | | 16 | 39 | 29.73 | 7.81 |
| Years of service as a nurse manager | | 2 | 26 | 12.81 | 7.94 |
| | | | | n | % |
| Gender | Women | | | 13 | 86.67% |
| | Men | | | 2 | 13.33% |

Min.–minimum, Max.–maximum, M–mean, SD–standard deviation

shortage of personal protective equipment, the staff resorted to making one on their own (e.g. masks).

> "*Some employees, as a protection, used diving masks with connectors from HEPA filters (High-Efficiency Particulate Air filters–author's explanation); we had to find a connector manufacturer on our own though.*" (WM2)

Despite the lack of personal protective equipment, nurse managers encountered situations where such items were misused (for example, filtered masks reserved for infected patients). As a result, nurse managers often had to ration and supervise the distribution of personal protective equipment or disinfectants among the staff.

Another challenge stemmed from the barriers associated with the architecture of hospitals and hospital wards. In order to put in place procedures governing the conduct of medical personnel during the pandemic, nurse managers had to reorganise their wards at a short notice. Among other things, they pointed out how poorly the infrastructure was adapted to the pandemic requirements. In consequence, they had to designate a special spot for the staff to change their clothes, accessible via separate (clean and dirty) routes. Many hospital wards did not have any isolation rooms, locks or separate entrances and exits. It was therefore the nurse managers' duty to ensure safe working conditions for both the staff and the patients, using the available means and resources.

> ". . .*It's just not possible for me to set up separation zones in my ward that would actually be effective. . . there's simply no room for them here. With its limited capacity, well, the ward is what it is. We won't move a wall or build anything, you know. . .*" (WM14)

Furthermore, nurse managers pointed to the problem of medical staff shortages as well as to the confusion resulting from insufficient information during the early days of the pandemic, e.g. on how employees should be quarantined and how they should return home. The nurse managers also felt that the reorganization of work during the COVID-19 pandemic–with rotations, teams split into groups, reassignment of duties–was a factor that aggravated divides in the team. Their nursing staff expressed hesitation about working with infected patients.

Another challenge which the nurse managers shared was the issue of how the young personnel would deal with the situation, especially with young mothers taking childcare leave, introduced by the authorities after schools and kindergartens were closed. Nurse managers talked also about the new organization of work, e.g. that due to staff shortages each nurse had to work one extra shift. ICU nurses–highly skilled professionals who can work everywhere– were expected to go to other wards to operate respirators.

> "*I can't simply 'clone' my nurses. . . I'm unable to provide them with normal working conditions.*" (WM14)

A significant problem for all nurse managers was the absence of clear and understandable procedures for dealing with, among other things, patients infected with SARS-CoV-2; this was accompanied by the staff's poor knowledge of and unwillingness to master particular procedures. The constant changes to the procedures in place further added to the confusion. Because of restrictions and limitations on human contact, documentation was mainly exchanged online, which posed another challenge for many nurses.

> "*All these procedures came up too late. It is only now that my colleagues are reading and signing them, and getting acquainted with everything.*" (WM4)

Since every patient was treated as potentially infected with SARS-CoV-2, nurses had to wear personal protective equipment. Putting on protective clothing initially took a lot of time, reducing the amount of direct patient care. The nurses performed their duties slower, with a limited field of vision, poorer sensation, and restricted movement–and they reported these issues to their nurse managers on a regular basis.

*New expectations of nursing workforce.* Some of the new expectations of nursing workforce, as reported by nurse managers, included a broader availability of tests for nurses and all hospital staff as well as allowances paid for work with SARS-CoV-2-infected patients. The staff also voiced the need for disposable clothing, setting up or providing rooms where they could bathe, rest or get some sleep, and architectural/infrastructure changes to their wards. As stressed by the respondents, however, relocating the walls or quickly restructuring the hospital space was not possible.

Nurse managers also strongly emphasised the constant need to be able to communicate up-to-date information to their subordinates. This included answering phone calls during the night to provide the required assistance and share instructions on what and how should be done, on a 24/7 basis.

"*Eventually, I decided there was no point for me to sit at home, keep stressing and look for answers to questions and requests... I came to the conclusion that I had to be there on the spot. You just can't do this job remotely, from home.*" (WM4)

*New roles of nurse managers.* A new role reported by most of the participants surveyed was that of the nurse manager as a therapist. The respondents noted the need to reassure and educate other employees so that they would not be afraid to perform their duties and would remain calm when providing patient care, as well as after returning home from work.

"*...And so I kept explaining to them it's just an infectious disease–because that's what it was, after all, maybe on a different scale. But basically it was nothing new... so we had no other choice than to reorganise and adapt to those new standards and recommendations that we received. And that's something we could do.*" (WM1)

An additional challenge for nurse managers was the labour-intensive nature of training on how to wear overalls, masks, visors, etc., as well as initial problems in convincing staff to use this equipment.

An important role and task mentioned by nurse managers involved activities that were part of their duties to "survive." They stressed the need to be able to respond to phone calls at all times during their shift. Equally important was their role in resolving conflicts and mitigating negative emotions among employees. Nurse managers also recounted numerous times when they had to show up for work outside of their regular shifts, putting their home responsibilities aside and giving up rest to provide assistance both to their subordinates and superiors.

"*We all do try to survive somehow*". (WM2)

"*It was my extra task to alleviate those negative feelings*". (WM3)

The nurse managers were also aware of their employees' family situation. They knew whether a particular nurse could leave her children by themselves, or whether she had parents who needed support and depended on her. In this way they could determine and decide which employees could, for example, come in to work for extra duty. All in all, nurse managers took on the roles of mentor, on the one hand, and of therapist, on the other.

"*I think that we all wanted that kind of mutual support, talking to each other more often, and a sense of reassurance that we indeed were needed; but also that we were scared, afraid for our families, our children. . ."* (WM11)

*Collaborating and dealing with different staff groups.* Nurse managers had different perceptions about their relations and cooperation with their superiors, physicians, and other nurses. Some of them pointed to the lack of support from hospital management, supervisors and medical personnel. They referred to their work during the pandemic as a 'lonely struggle'.

"*Working together, as part of collaboration between the doctors, the staff, and the hospital principal, has never been perfect and I think it will remain so. . . There's nothing [we] have learned from this experience".* (WM14).

Nurse managers also expressed their dissatisfaction with hospital management's decisions which were to be communicated to other employees and which would further aggravate an already difficult situation in hospital wards.

"*So what if I tell someone that I have no support here and that I feel like left out alone on the front line. It won't change anything because there's no willingness to change things here. So all this complaining, so to speak, won't do much".* (WM1)

A major barrier stressed by the respondents was the lack of proper communication between nurses and doctors. The COVID-19 pandemic has, in many cases, as the nurse managers described it, exacerbated relations and conflicts between the members of various medical professions.

"*And then there was this conflict between the doctors and the nurses. I think it got even worse. . . ethical or not, but it did get worse".* (WM14)

Some of the nurse managers talked about how the physicians became withdrawn during the pandemic, how they reduced direct contact with patients and would only write down treatment recommendations, preferring to be contacted by phone, if at all. Overall, they showed impatience and limited understanding for both the patients and other members of the treatment team. Also noted by the nurse managers was their inability to influence the physicians, e.g. by encouraging them to help nurses more and work with the patients during the pandemic, as expected by their subordinates.

"*The doctors never helped us when, for example, the patient had to be moved on bed–they couldn't be bothered. So expecting me to, I don't know, make them work with the patient, to change something. . . Everyone knows how it is. So I won't change that, I just can't do it. Plus the problems with the equipment, means of protection, rotas. . . and on top of that I need to act as a therapist".* (WM14)

That withdrawing, however, could be observed among the nursing staff as well. Nurse managers noted difficulties with communication inside the team, in addition to a sort of stigma put on those staff members who had a more frequent contact with SARS-CoV-2 patients, and nurses' dissatisfaction with changes and reassignment of their duties, and with the new systems of work. Other medical professions tried to minimise the risk of infection by limiting the tasks they had previously performed on a regular basis. And so, for example, nurses had to collect

laboratory test results on their own, which obstructed their work and caused conflicts that the nurse managers had to resolve.

> "*Everyone sort of tries to self-isolate and minimise the risk.*" (WM1)

On the other hand, however, nurse managers observed greater consolidation among the nursing staff as manifested through mutual support and assistance or understanding and communicating without words. By working together, nurses and physicians tightened their bonds and could understand each other better. The respondents observed that nurses working hand in hand with physicians for 3 hours, attending the patients, and having to stay together in the patient room brought about more openness, humility, and tolerance.

> "*I think that, at the personal level, we became more tolerant of each other.*" (WM10)

The nurse managers also described various relationships with District Sanitary and Epidemiological Stations and nurse epidemiologists. In this regard, both positive aspects were mentioned, with nurse managers able to effectively cooperate with nurse epidemiologists and receive the necessary help and advice from them, as well as negative ones. The latter included unwillingness to cooperate, unavailability of clear guidelines and the fact that any difficult decisions were left to the nurse managers.

*Emotions and a sense of responsibility for the team.* The prevailing emotions experienced by the nurse managers were as follows: (1) fear and panic at the beginning, but then the attitude of "the more we know and the more equipment we have, it gets better"; (2) concerns regarding the personnel, i.e. whether they correctly use PPE and are generally well protected; (3) concerns related to the staff's negligent approach to safeguarding and exposing other members of the treatment team to the risk of infection.

> "*I'm especially worried that something happens to them, that something goes wrong and our team will totally fall apart*". (WM4)

Other emotions frequently described by nurse managers were fear and a very strong sense of responsibility for the team (employees' health, breakdown of relations, conflicts, procedures being incorrectly followed, compromising patient health). Instead of working from home, many of the respondents preferred to lead their teams in person, in the ward. This allowed them to quickly help their subordinates in crisis situations. Furthermore, they would often choose to complete various tasks themselves to ensure that they were performed correctly and to have the feeling that everything was done as it should be and the team was safe. Nurse managers also reported stress or reluctance when answering phone calls while they were at home, especially from other staff members during the night. The constant need to be available led to inability to relax.

Exhaustion and fatigue made both nurse managers, nurses and physicians more nervous. Whenever one person was panicking, others soon joined. Nurse managers recounted situations in which they cried after returning home due to helplessness and flood of emotions. Others had memory lapses from the excess of information and sensory overload. Several respondents said that the crisis situation triggered a motivation to act.

> "*Well, there is this a quite heavy... mental burden... I remember laughingly saying that once it is all over, I will end up with PTSD. I laughed then, but now I'm actually afraid, because I'm already having trouble sleeping, for example (...) I have memory lapses from this excess*

*of information and sensory overload and all these nerves and emotions. . . but it suits me".* (WM3)

Some of the nurse managers expressed regret over the failure of the hospital management, the head nurse, to take advantage of their experience. They lacked the ability to analyse the situation together and look for solutions. Another group mentioned a sense of fulfillment derived from working as a nurse manager during the pandemic.

"*We were just making the world a better place from scratch (. . .) And I did have a sense of accomplishment". (WM13)*

*Current ethical dilemmas.* Nurse managers noted that nurses began to spend less time with patients. They reduced their patient-related work to a minimum, which resulted in less than the highest level of care. The patients were sort of forgotten.

"*We enter the patients' room, do our job, and leave*". (WM2)

Another dilemma for nurse managers revolved around the allocation of nursing staff to work with infected patients. They had to decide whether to assign nurses from risk groups, i.e. those with young children or older nurses with chronic illnesses, to tasks with potential exposure to infection.

They also wondered how to proceed in case of equipment shortages (masks, respirators, etc.), to whom to allocate the equipment, and at what point to stop futile treatment. According to the respondents, the COVID-19 pandemic also proved a testing point for the motivation to become a nurse. The difficult situation and work under stressful conditions revealed to the nurse managers who had made an informed choice to purse this career and who chose it by accident or for money.

"*Some do want to help while others will do anything to escape the responsibility–I guess it's out of fear*". (WM2)

The nurse managers also reported that they were unable to ensure the observance of patients' rights, including the right to contact family. In some cases, patients were being prevented from saying their last goodbyes to their loved ones. Despite the nurse managers' requests and readiness to provide the required safety measures, consent for a visit was often not given.

"*They cry and I cry but that's all since I can't help them anyway. . .*" (WM4)

**Nurse managers' expectations.**    *Urgent needs.* The nurse managers expressed the need for more support from their superiors, including help in setting up workstations for the nurses. In many cases, it was the nurse manager who was responsible for arranging and providing the needed equipment. Hospital executives were unavailable and did not provide the advice and assistance that was expected of them.

"*In my view the executives should be providing support to those lower on the management ladder, and us would be supporting nurses on the front line. At least that's how I see it".* (NM9)

Another expectation mentioned by nurse managers was to devise top-down clear, transparent, and well-thought-out routes and procedures. Detailed information and procedures were

not available until 2 months after the pandemic started. Initially, this situation had a substantial negative impact. Materials, translated from Chinese, along with media images formed an overall picture that fueled fear. The respondents also reported feelings of hope and anticipation for a vaccine to be invented and deployed, knowing that without it the COVID-19 pandemic would not stop.

Furthermore, the respondents pointed to the need for psychological support among nurses and nurse managers. Some nurse managers reported how their subordinates were unwilling to talk to the hospital psychologist because they did not trust him or her. They were afraid that the entire ward or even hospital community would learn what they had said in secret to the psychologist.

*Thinking about the future.* Nurse managers declared readiness to continue using certain PPE items (masks, visors, etc.) in the future and to incorporate them into standards of conduct and medical procedures. They would also like medical staff to maintain the established level of disinfection care and hand hygiene.

"*It's noticeable among nurses, you know, this greater care of self-protection. . . but also protecting the patients from potential infection. . . there is greater emphasis on hand hygiene. . . The people themselves, without being told so, have given up on jewelry. . . They also stopped painting nails, which previously was pretty much brushed off. But now definitely some progress could be seen here. This and disinfection–I can see that disinfection, along with hand hygiene, have gained in importance*". (WM6)

When asked about the future, nurse managers expressed the need for onsite training in epidemiology, pandemics, crisis management, or ability to anticipate specific events. In addition to training, architectural changes were also mentioned: remodeling hospital space, creating isolation rooms, locks, and non-overlapping clean and dirty routes, and providing newer and better equipment than what they currently have at their disposal.

When asked about the future, nurse managers considered leaving employment during the national quarantine. However, for most of them it was more important to take care of their teams, rebuild relationships, hire more medical staff and train them on a regular basis, secure the necessary resources, and develop inventory. Nurse managers recounted the common dream of keeping their subordinates and their families healthy. They also expressed a desire to return to normality, although they still had doubts about whether the COVID-19 pandemic would end.

**Methods of coping and received support.** Some of the coping methods listed by the respondents included spending time outdoors with their families, reading books, walking or cycling, doing yoga, or taking a relaxing bath. An important aspect here was the support from their families and the relationships within the treatment team, especially between the nurse managers and their subordinates. Talking and laughing together, taking a break or having a conversation with a colleague or trusted psychologist served as an outlet for any negative emotions that might have accumulated.

"*If something's bothering you, the most important thing is to talk to someone. And if you keep it to yourself, you're in for headaches, chest pain, sleep problems and whatnot. . .*" (WM11)

"*Even helping each other dress and undress brought us closer, almost like in a family. . .*" (WM12)

Also helpful and supportive to some of the nurse managers were their supervisors. Procedures, up-to-date information and availability via email, video calls or support from good-

willed people proved an effective source of mental comfort. All equipment was available in hospitals dealing with Covid patients, and where any shortage was reported, it was immediately dealt with by senior management. They even provided food and drinks for the employees who were generally well catered for. Another important aspect was the mutual support between junior and senior colleagues. This was in addition to aspects such as setting an example for others, education, staying calm in difficult situations, as well as training, online webinars, and obtaining information from, e.g. the Association of Anesthesia and Intensive Care Nurses.

> "*Knowing that my ward is in good order and everything is organised as it should be–that's what gives me peace of mind". (WM9)*

On the other hand, some nurse managers indicated that it was not possible for them nor were they capable of relaxing and coping with the difficult, new situation of managing the ward and employees during the COVID-19 pandemic. One of the methods of coping with stress and negative emotions was to take so called fetal position when laying down or sleeping (bowing the head forward, arching the back, bending at the hips, and flexing at the knees). Many respondents also talked about not having enough strength to engage in activities such as reading or watching TV.

> "*When coming back home I only wish to get some rest. . . And that nobody calls me from work. . . That everything is OK there". (WM2)*

> "*I'm too tired to read a book, I don't even know what I'm reading. Television isn't interesting either. I just lie down on the couch and try to unwind". (WM3)*

## Discussion

The purpose of the study was twofold: (1) to explore the experiences of the ICU nurse managers regarding their work during the first wave of the COVID-19 pandemic, and (2) to analyse what implications might be provided based on experiences of nurse managers for future possible epidemiological crises. The fight with the coronavirus pandemic would have been impossible without the brave and heroic attitude of health care personnel, and nurses in particular. In addition, this difficult situation highlighted the problems faced by nurse managers. Literature describes the main conundrums which hindered the functioning and fulfilment of professional duties by nurses [20, 21, 28–32].

As regards the first category that emerged from the analysed interviews, all of the study participants unanimously confirmed that it was a difficult time for them. Most of the nurse managers interviewed were not prepared to manage the COVID-19 pandemic from its start. At first there was a lack of guidelines on how to act in this new reality, and then it was difficult to keep up. The total reorganization of work–the need to equip staff working in the ward with personal protective equipment, providing disinfectants, managing staff rotations and shortages, arranging additional on-call duties, solving the problem of outdated medical equipment–it all posed challenges of ethical nature. The same findings have been reached by other researchers [28, 33], who indicate that this kind of situations exerts heavy, both physical and mental, burden. Undoubtedly, as Vázquez-Calatayud et al. point out, the constant ability to adapt quickly to change, to participate in decision-making processes in the organizational unit in a variety of positions, to overcome the uncertainty of the situation, to set priorities for the biopsychosocial wellbeing of staff, to ensure that the humanitarian principles of care are

preserved, as well as to create opportunities for teamwork and multi- and interprofessional cooperation is a heavy burden for the team management personnel [31].

Nurse managers pointed out that it is slower to work around patients when wearing protective uniforms, due to limited vision, sensing and movement. Severe working conditions (staying in a protective suit for 8 hours was impossible, face masks caused abrasions of the facial epidermis, and since route separation was preserved, there was no place where one could change freely [6, 34]. Therefore, it is necessary to educate nursing staff on strategies regarding the use of personal protective equipment so that the consequences of the pandemic are as small as possible [9].

This unique situation has imposed new challenges among nurse managers, as they must handle difficult ethical decisions under the pressure of a pandemic environment, distributing limited resources among staff, and taking responsibility for the team [35]. In Gordon et al. study, nurses indicated that the greatest challenge for them during the pandemic was to deal with the deaths of their patients, day in and day out, allow patient visits due to restrictions, keeping pervasive social distance and necessary isolation, and using limited personal protective equipment [36]. Nurse managers complained of there being more documentation, procedures, and referrals for swab tests to be filled out. Interestingly, the introduction of electronic medical records optimization into the program is an effective solution and contributes to an increase in nurses' involvement at the bedside [37].

A major challenge brought about by the outbreak of the pandemic was to organise nursing teams, as there was a shortage of workforce in this area. It was difficult to arrange schedules, as many nurses were on a sick leave or on a parental leave because they had to care for a child. The reasons for this can be found in excessive workloads due to work at several places, along with nurses' occupational burnout, fear for their own health and life, and fear of infecting their family members [12, 38–41]. In Poland, the average age of a nurse is over 53 [42], so it is necessary to encourage young people to study nursing in order to close the gap in the nursing workforce. As for interprofessional cooperation, the work situation was tense and stressful, and this may have been due to heavy mental strain and fatigue [43]. A good solution would be to introduce classes in university curricula where different medical professions could work together as an interdisciplinary team in patient care. In addition, the communication and flow of information is also a protective factor against uncertainty and feelings of anxiety [44, 45].

The COVID-19 pandemic has increased the incidence of ethical problems in the delivery of care by nurses [46]. First, the patient was relegated to the background, with nurses performing only their basic tasks, very quickly, and then leaving the infected space. Moreover, the limited amount of medical equipment (such as ventilators) made it necessary to choose which patient should use it, thus putting other patients aside. Here, time was of the essence and there was no room for procrastination. This may sound harsh, but can be justified in the reorientation of nursing, which so far has been person-centered, but has shifted to focus more on the health needs of general population [47]. Many researchers stress the need to develop a basic framework for dealing with a similar situation, as this will avoid disorganization caused by not knowing how to behave [48, 49].

The pandemic has left its mark on mental health among nursing staff [50–54]. The constant uncertainty of whether an effective vaccine against SARS-CoV-2 would be invented stirred up negative emotions such as fear, anxiety, and worry about the future. Nurse managers felt that they were particularly responsible for their team, worried about the well-being of their co-workers so that they would persevere until the end of the pandemic, and so they played a dual role: guardians of the health safety of their subordinates and patients, and motivators of the nursing staff encouraging them to provide comprehensive care [55]. Working as a nurse has become more stressful for nurse managers. It has been always stressful at ICU but now it has

only aggravated. This opens a new chapter in the leadership of nurses, making it transformational leadership, due to the need to respond quickly to the dynamically changing conditions of the work environment [56].

The main strategy for coping with stress among nurse executives was to avoid and drive away negative thoughts [57]. In order to relax, respondents took a relaxing bath, spent time with their families outdoors, read books, undertook physical activity such as yoga or cycling. While at work they mainly talked, joked and laughed. However, professional help, e.g. from a psychologist is needed, as lack of this type of support can contribute to mental disorders such as PTSD (Post-Traumatic Stress Disorder) [58].

The respondents' main source of support was family. They could also count on support coming from within the organization, including hospital management, and encouragement from each other, between junior and senior colleagues. With access to multidimensional support, nurse managers experienced positive emotions [34, 59].

One of the basic elements that make up the support dimension are the financial and verbal rewards given by management [60]. For working under difficult and strenuous conditions, nurse managers and their teams received so-called "Covid allowances", which certainly compensated, at least in part, any inconveniences and helped encourage many nurses to stay on the job [61].

## Conclusions

The COVID-19 pandemic forced ICU nurses into new roles and responsibilities, and exposed the malfunctioning of the healthcare system, that had been ignored for many years. The shortages of nursing personnel, so manifested before the pandemic, critically exacerbated due to frequent infections among nurses, sick leaves and quarantines, and parental leaves. This made it difficult for nurse managers to ensure full staffing during on-call periods, while being one of the main reasons for the stress they experienced. The mental burden on ICU nurse managers was associated not only with staffing, but also with the sense of responsibility for the safety of their nursing teams.

The experience of ICU nurse managers has shown how important it is to think ahead and prepare in advance for potential epidemiological crises which regularly occur around the world. It is worth keeping them in mind as early as at the stage of planning the construction of a hospital and furnishing of the wards, as well as in the process of developing ICU procedures, and channels of communication between various professions and management levels.

It is vital that in-service training be organised for ICU nursing management personnel on how to deal with emergencies, during the pandemic, in order to manage healthcare facilities more effectively, and to increase patient and staff safety and the quality of patient care.

## Strength and limitations of the study

The study has several limitations. We interviewed the participants at the beginning of the pandemic, therefore at a time when the outbreak was still emerging and little known. Furthermore, at this stage the participants were not personally and emotionally involved in the severity that was reported in the subsequent waves. However, the study captured the first phase of the ICU involvement, at an early stage, giving suggestions on how this can be managed. The data was immediately analyzed to identify the saturation; however, their interpretation benefited from the contributions that subsequently emerged from the literature, thus also in light of the overall experience, evidence produced and the progressive awareness of the situation.

## Supporting information

**S1 File. COREQ (Consolidated criteria for Reporting Qualitative research) checklist.**
(DOCX)

## Acknowledgments

The authors would like to thank all nurse managers who agreed to take part in the study. Even though the situation in the first half of 2020 was extremely difficult they found the time to share their experiences in order to draw conclusions and create comprehensible guidelines for the future epidemiological crises.

## Author Contributions

**Conceptualization:** Beata Dobrowolska, Aleksandra Gutysz-Wojnicka, Magdalena Dziurka, Patrycja Ozdoba, Dorota Ozga, Beata Penar-Zadarko, Renata Markiewicz, Agnieszka Markiewicz-Gospodarek, Alvisa Palese.

**Data curation:** Beata Dobrowolska, Aleksandra Gutysz-Wojnicka, Magdalena Dziurka, Patrycja Ozdoba, Dorota Ozga, Beata Penar-Zadarko, Renata Markiewicz, Agnieszka Markiewicz-Gospodarek.

**Formal analysis:** Beata Dobrowolska, Magdalena Dziurka, Patrycja Ozdoba.

**Investigation:** Beata Dobrowolska.

**Methodology:** Beata Dobrowolska, Aleksandra Gutysz-Wojnicka, Magdalena Dziurka, Patrycja Ozdoba, Dorota Ozga, Beata Penar-Zadarko, Renata Markiewicz, Agnieszka Markiewicz-Gospodarek, Alvisa Palese.

**Supervision:** Beata Dobrowolska, Alvisa Palese.

**Visualization:** Beata Dobrowolska, Magdalena Dziurka, Patrycja Ozdoba, Beata Penar-Zadarko, Renata Markiewicz, Agnieszka Markiewicz-Gospodarek, Alvisa Palese.

**Writing – original draft:** Beata Dobrowolska, Aleksandra Gutysz-Wojnicka, Magdalena Dziurka, Patrycja Ozdoba, Dorota Ozga, Beata Penar-Zadarko, Renata Markiewicz, Agnieszka Markiewicz-Gospodarek, Alvisa Palese.

**Writing – review & editing:** Beata Dobrowolska, Alvisa Palese.

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
