## [Decision Letter · Decision Letter 0]

18 Jun 2023

PONE-D-23-12373Steering a boat in a storm. Qualitative analysis of ICU nurse managers’ experiences during the first wave of the Covid-19 pandemic: implications for future epidemiological crisesPLOS ONE

Dear Dr. dobrowolska,

Thank you for submitting your manuscript to PLOS ONE. After careful consideration, we feel that it has merit but does not fully meet PLOS ONE’s publication criteria as it currently stands. Therefore, we invite you to submit a revised version of the manuscript that addresses the points raised during the review process.

Shorten the Title and do not use abbreviations. Specify the number of participants as well as the type of study design used. 

We look forward to receiving your revised manuscript.

Kind regards,

Sogo France Matlala, PhD

Academic Editor

PLOS ONE

2. In the ethics statement in the Methods, you have specified that verbal consent was obtained. Please provide additional details regarding how this consent was documented and witnessed, and state whether this was approved by the IRB

Reviewers' comments:

Reviewer's Responses to Questions

**Comments to the Author**

1. Is the manuscript technically sound, and do the data support the conclusions?

Reviewer #1: Yes

Reviewer #2: Yes

2. Has the statistical analysis been performed appropriately and rigorously? 

Reviewer #1: Yes

Reviewer #2: Yes

3. Have the authors made all data underlying the findings in their manuscript fully available?

Reviewer #1: Yes

Reviewer #2: Yes

4. Is the manuscript presented in an intelligible fashion and written in standard English?

Reviewer #1: Yes

Reviewer #2: Yes

5. Review Comments to the Author

Reviewer #1: The manuscript authors have followed the PLOS authors guidelines. The manuscript will add to the body of knowledge about the impact of COVID 19 pandemic on the nursing profession. It is well written and the both data collection analysis procedures are relevant.

Reviewer #2: Line 34 Suggest align number of interviewed nurse managers with the number indicated in methodology as in line 111 and 112 below is indicated that theoretical saturation achieved at 18 participants for consistency and clarity of the exact numbers where data saturation was achieved

Line 41 Suggest the training be’specific for “an in-service training” not merely training

Line 98 Suggest the study design is clearly reflected in this section and rationale for choosing the design

Line 102 Recheck this under measures to ensure trustworthiness in qualitative research , suggest use of dependability as reliability is more of a quantitative approach

Line 130 Suggest indicate clearly how this triangulation was done to improve clarity

Line 158 Suggest this number be clarified as in line 112 saturation was achieved at 18 participants

Line 403 Suggest be consistent with use of terms, of where is referred to as subjects, nurse managers, and respondents

Line 463 Suggest rephrasing of statement to improve meaning

6. PLOS authors have the option to publish the peer review history of their article (what does this mean?). If published, this will include your full peer review and any attached files.

Reviewer #1: No

Reviewer #2: **Yes: **Mutshatshi Takalani Edith

---

## [Author Response · Author response to Decision Letter 0]

24 Jul 2023

Response to Editor

Dear Editor,

Thank you very much for your very informative feedback and a given chance to correct our manuscript. We have carefully implemented all suggestions indicated by you and also by reviewers. All changes are introduced in a form of ‘track changes’ and below explanation is written what was changed.

Shorten the Title and do not use abbreviations. Specify the number of participants as well as the type of study design used. 

Thank you for this important remark. We have shorten the title of the manuscript. The number of participants in the study is specified in abstract and in the manuscript body, and also the study design.

Thank you for this notice. We changed file naming according to journal requirement. 

2. In the ethics statement in the Methods, you have specified that verbal consent was obtained. Please provide additional details regarding how this consent was documented and witnessed, and state whether this was approved by the IRB

Thank you for this important remark. We added detailed information about obtaining verbal consent from study participants and how it was documented. Additionally, we also added that the protocol of the study accepted by IRB contained detailed information about the process of obtaining verbal consent.

Thank you for this notice and apologies for misunderstanding. We did not receive any funding for this study, therefore we stated in a cover letter: “The authors received no specific funding for this work.”

4. In your Data Availability statement, you have not specified where the minimal data set underlying the results described in your manuscript can be found. 

Thank you for this notice. We have now explained this issue in a cover letter.

5. Please review your reference list to ensure that it is complete and correct.

Thank you for this information. We checked our reference list. During revision we added one additional reference which is indicated in the manuscript body.

Response to Reviewers

Reviewer #1: 

Dear Reviewer, thank you very much for the review of our manuscript and your encouraging comments! 

The manuscript authors have followed the PLOS authors guidelines. The manuscript will add to the body of knowledge about the impact of COVID 19 pandemic on the nursing profession. It is well written and the both data collection analysis procedures are relevant.

Reviewer #2: 

Dear Reviewer, thank you very much for valuable feedback that we received, which helped us to improve our manuscript. We corrected our work strictly following your suggestions. All changes are provided in a form of ‘track changes’ and are visible in a manuscript body. Below we also included point-by-point answer to all your remarks.

Line 34 Suggest align number of interviewed nurse managers with the number indicated in methodology as in line 111 and 112 below is indicated that theoretical saturation achieved at 18 participants for consistency and clarity of the exact numbers where data saturation was achieved

Thank you for this remark. We would like to apology for not being clear in stating the number of interviews with nurse managers included in final analysis. It is now corrected.

Line 41 Suggest the training be’specific for “an in-service training” not merely training

Thank you. We have specified this issue as suggested.

Line 98 Suggest the study design is clearly reflected in this section and rationale for choosing the design

Thank you for this notice. We included clear information about the study design and also justification why this study design was chosen.

Line 102 Recheck this under measures to ensure trustworthiness in qualitative research , suggest use of dependability as reliability is more of a quantitative approach

Thank you very much for this remark. We have corrected reliability into dependability as it should be.

Line 130 Suggest indicate clearly how this triangulation was done to improve clarity

Thank you for this notice. We have explained how triangulation was done in more detail.

Line 158 Suggest this number be clarified as in line 112 saturation was achieved at 18 participants

Thank you. We have corrected the number (n=15).

Line 403 Suggest be consistent with use of terms, of where is referred to as subjects, nurse managers, and respondents

Thank you. We corrected everywhere into nurse managers.

Line 463 Suggest rephrasing of statement to improve meaning

Thank you. We corrected indicated sentence.

Thank you once again for your valuable suggestions to improve our manuscript.

---

## [Editor Report · Decision Letter 1]

13 Aug 2023

Intensive care nurse managers’ experiences during the first wave of the Covid-19 pandemic: implications for future epidemiological crises

PONE-D-23-12373R1

Dear Dr. dobrowolska,

We’re pleased to inform you that your manuscript has been judged scientifically suitable for publication and will be formally accepted for publication once it meets all outstanding technical requirements.

Kind regards,

Sogo France Matlala, PhD

Academic Editor

PLOS ONE
---

## [Editor Report · Acceptance letter]

18 Aug 2023

PONE-D-23-12373R1 

Intensive care nurse managers’ experiences during the first wave of the Covid-19 pandemic: implications for future epidemiological crises 

Dear Dr. Dobrowolska:

I'm pleased to inform you that your manuscript has been deemed suitable for publication in PLOS ONE. Congratulations! Your manuscript is now with our production department. 

Kind regards, 

on behalf of

Professor Sogo France Matlala 

Academic Editor

PLOS ONE